# Selectively exciting quasi-normal modes in open disordered systems

Matthieu Davy[1] & Azriel Z. Genack[2]

Transmission through disordered samples can be controlled by illuminating a sample with waveforms corresponding to the eigenchannels of the transmission matrix (TM). But can the TM be exploited to selectively excite quasi-normal modes and so control the spatial profile and dwell time inside the medium? We show in microwave and numerical studies that spectra of the TM can be analyzed into modal transmission matrices of rank unity. This makes it possible to enhance the energy within a sample by a factor equal to the number of channels. Limits to modal selectivity arise, however, from correlation in the speckle patterns of neighboring modes. In accord with an effective Hamiltonian model, the degree of modal speckle correlation grows with increasing modal spectral overlap and non-orthogonality of the modes of non-Hermitian systems. This is observed when the coupling of a sample to its surroundings increases, as in the crossover from localized to diffusive waves.

---

[1] Institut d'Electronique et de Télécommunications de Rennes, University of Rennes 1, 35042 Rennes, France. [2] Department of Physics, Queens College and Graduate Center of the City University of New York, Flushing, NY 11367, USA. Correspondence and requests for materials should be addressed to A.Z.G. (email: genack@qc.edu)

Transmission of a monochromatic wave through a static sample is fully described by the transmission matrix (TM), **t**. The TM is a subset of the scattering matrix which provides the coupling of a system to its surroundings. In the last decade, there has been growing interest in measuring the TM to control the flow of waves through random systems and optical fibers[1,2]. Shaping the incident waveform illuminating a sample makes it possible to manipulate the net transmitted flux and its spatial intensity profile for applications in medical imaging and communications. For instance, a scattering medium can appear to be transparent or opaque when the incoming wavefront is adjusted to correspond to the first or last transmission eigenchannels[3–9]. The intensity can also be focused through random media at a selected point in the output by adjusting the incident wave so that all transmission channels interfere constructively at that point[1,10].

The elements $t_{ba}$ of the TM are the field transmission coefficients between the $N$ channels leading toward and away from opposite ends of a sample, $a$ and $b$, respectively. The TM was initially studied in order to explain the scaling of the conductance of wires at zero temperature[11,12]. Classical and quantum transport are connected by the dimensionless conductance, which is the conductance in units of the quantum of conductance, $(e^2/h)$. The dimensionless conductance is equal to the average transmittance, $g = \langle T \rangle$, where $\langle \cdots \rangle$ represents the average over a random ensemble. The crossover to Anderson localization occurs at $g = 1$; waves are localized for $g < 1$ and diffusive for $1 < g < N/2$. The transmittance is the sum of all flux transmission coefficients, $|t_{ba}|^2$, which equals the sum of the $N$ eigenvalues $\tau_n$, $T = \Sigma_{a,b=1}^{N}|t_{ba}|^2 = tr(\mathbf{tt}^\dagger) = \Sigma_{n=1}^{N}\tau_n$[11,12].

In principle, the degree of control over transmission in diffusive samples is strong because the distribution of transmission eigenvalues is wide. This distribution is bimodal with a peak near unity containing $g$ "open" channels and a second peak corresponding to "closed" channels with values that are exponentially small in the ratio of the sample length and the transport mean free path, $L/\ell$[6,12–14]. In practice, however, measurements of the TM are incomplete so that the dynamic range over which transmission can be controlled is limited[4,15–17]. Because the dwell time and the energy density profile inside a medium excited in a specific eigenchannel are correlated with the corresponding transmission eigenvalue, exciting transmission eigenchannels also provides a measure of control over the dwell time and the spatial distribution of energy within a random medium[3,5–9,18]. The full diversity of dwell times is given by the eigenvalues of the Wigner-Smith time-delay matrix, known as the proper delay times, which are constructed from the spectrum of the scattering matrix[19–22].

Another approach to controlling propagation within random or structured media might be to manipulate the incident wave to preferentially excite specific quasi-normal modes[23] which have different lifetimes and spatial profiles. Modes of open systems are solutions of the wave equation over the volume of the random medium with outgoing radiation boundary conditions[24–26]. In resonating structures for which the complex eigenvalues and eigenvectors can be found analytically or numerically, the field for any source excitation can be reconstructed from the coherent superposition of modal contributions[26,27]. Beyond the independent contribution of each mode, the resultant field depends critically upon the interference between the fields of modes that overlap spectrally and spatially. Modal coupling plays a key role in describing the physics of photonic systems such as chaotic cavities[28–32], coupled cavities or waveguides[33,34], optical resonators[27,35], quantum plasmonic[36,37] or disordered media[30,38–40].

In large complex systems, it is generally not possible to solve for the eigenvectors of the wave equation, but important properties of a system and its coupling to its surroundings can be determined from the statistics of scattering spectra and their analysis into modes or energy levels. Great emphasis has been placed on the probability distributions of level spacings[41–43] and level widths[28,32,44]. However, the statistics of level widths and spacings do not directly yield the statistics of scattering because the scattered wave also reflects the interference between modes and the degree to which modal speckle patterns are correlated.

Here we consider the degree of modal selectivity that can be achieved in random media by manipulating the incident waveform. We approach the problem by analyzing the spectrum of the TM into its modal components in locally 2D $N$–channel samples. The complex modal frequencies and amplitudes are found by decomposing the spectra of the elements of the TM into a superposition of spectral lines via Breit-Wigner theory[30,45,46]

$$t_{ba}(\omega) = \Sigma_n \frac{t_{ba}^n}{\omega - \omega_n + i\Gamma_n/2} = \Sigma_n t_{ba}^n \varphi_n(\omega). \tag{1}$$

Here $\varphi_n(\omega) = (\omega - \omega_n + i\Gamma_n/2)^{-1}$ is the frequency variation of excitation of the field associated with the mode with central frequency $\omega_n$ and linewidth $\Gamma_n$, and $t_{ba}^n$ is the complex field transmission coefficient associated with the $n$th resonance. Each resonance is then associated with a modal transmission matrix (MTM), $t_n$, which is built upon the coefficients $t_{ba}^n$, and is the contribution of a mode of the scattering medium to the TM[23]. An MTM therefore provides the incoming wavefront that maximally enhances the energy in a specific mode. However, modal selectivity becomes more challenging as the degree of modal overlap increases in non-Hermitian media.

## Results

**Modal decomposition in the effective Hamiltonian formalism.** The coupling of a system to its surroundings can be analyzed in terms of an effective Hamiltonian. The coupling is described via the $2N \times 2N$ scattering matrix $S$ expressed in terms of the $M \times M$ effective Hamiltonian $\mathbf{H}_{\text{eff}}$ as[28–30,47–49]

$$\mathbf{S} = \mathbf{1} - i\mathbf{V}^T \frac{1}{\omega - \mathbf{H}_{\text{eff}}} \mathbf{V}. \tag{2}$$

Here $\mathbf{V}$ is a real $M \times 2N$ matrix describing the coupling of the $M$ modes of the closed system to the exterior via the $2N$ channels in the leads on both sides of the sample. The non-Hermitian effective Hamiltonian is

$$\mathbf{H}_{\text{eff}} = \mathbf{H}_0 - \frac{i}{2}\mathbf{V}\mathbf{V}^{\mathbf{T}}, \tag{3}$$

where $\mathbf{H}_0$ is the Hermitian Hamiltonian of the closed system. The poles of the $\mathbf{S}$ matrix occur at the complex eigenvalues $\tilde{\omega}_n$ of $\mathbf{H}_{\text{eff}}$, $\tilde{\omega}_n = \omega_n - i\Gamma_n/2$. Two sets of eigenvectors are associated with these eigenvalues. These are the right $|\phi_n\rangle$ and left $\langle\varphi_n|$ eigenvectors, which are the transpose of one another, $\langle\varphi_n| = (|\phi_n\rangle)^T$. The eigenfunctions of $\mathbf{H}_{\text{eff}}$ are bi-orthogonal and satisfy the following orthogonality condition due to the time-reversal symmetry of $\mathbf{H}_{\text{eff}}$, $\langle\phi_n^*|\phi_m\rangle = \delta_{nm}$. As a result of the imaginary part of $\mathbf{H}_{\text{eff}}$, the eigenfunctions are complex.

The modal decomposition of the TM may be expressed in terms of the complex vectors $|\mathbf{W}_{Ln}|^2$ and $|\mathbf{W}_{Rn}|^2$ which couple the eigenstates $\varphi_n$ to the scattering wavefunctions $\xi_L$ and $\xi_R$ in the left

and right leads, respectively[29],

$$\mathbf{t}(\omega) = -i \sum_{n=1}^{M} \frac{\mathbf{W}_{Rn}\mathbf{W}_{Ln}^{T}}{\omega - \omega_n + i\frac{\Gamma_n}{2}}. \qquad (4)$$

We identify the MTM of the $n$th mode at resonance as $\mathbf{t}_n = -i\mathbf{W}_{Rn}\mathbf{W}_{Ln}^{T}/(\Gamma_n/2)$. The MTM for each mode is of unit rank since it is the product of the vectors $\mathbf{W}_{Rn}$ and $\mathbf{W}_{Ln}^{T}$. In principle, the coupling of the eigenfunctions of the closed system to the leads, and therefore vectors $\mathbf{W}_{Rn}$ and $\mathbf{W}_{Ln}$, depend on frequency. However, in the case of resonances with high quality factors $Q_n = 2\omega_n/\Gamma_n$, which are explored in the experiments described below, we can take $\mathbf{W}_{Rn}(\omega) = \mathbf{W}_{Rn}(\omega_n) = \mathbf{W}_{Rn}$ and $\mathbf{W}_{Ln}(\omega) = \mathbf{W}_{Ln}(\omega_n) = \mathbf{W}_{Ln}$. The TM is then expressed as a superposition of MTMs with modal transmission coefficients for Lorentzian lines defined at the resonance frequency. The decomposition of the TM into MTMs and the properties of the MTMs are demonstrated below in microwave experiments.

**Experimental setup.** Measurements of the TM are performed in a two-dimensional cavity containing randomly positioned disks

(see Fig. 1a and Methods for details). Spectra of the $N \times N$ TM are measured between two arrays of $N = 8$ emitting and receiving antennas on the left and right sides of the cavity, respectively. Measurements are carried out in the frequency range 10.7–11.7 GHz in a scattering sample of 300 randomly distributed 6-mm-diameter aluminum disks. The sample is weakly localized and modal spectral overlap is moderate. The coupling strength $\tilde{T}_a$ of the antennas to the sample is determined using the mean value of the reflection parameter at each antenna, $\langle S_{aa} \rangle$, $\tilde{T}_a = 1 - |\langle S_{aa} \rangle|^2$ gives $\tilde{T}_a \sim 0.99$ so that the antennas are strongly coupled to the cavity.

**Decomposition into MTMs.** The modes can be found from an analysis of the spectrum of the TM as a superposition of MTMs using Eq. (1). The set $\{\omega_n, \Gamma_n\}$ is extracted via the Harmonic inversion (HI) method from the inverse Fourier transform of spectra of transmission coefficients $t_{ba}(\omega)$[45,50,51] (see Methods). The coefficients $t_{ba}^n$ of $t_n$ are then found from the fit of transmission coefficients in the time domain. The flux transmission coefficient between two channels, $|t_{ba}(\omega)|^2$, and the underlying modal transmission coefficients between the channels for each mode, $|t_{ba}^n|^2|\varphi_n(\omega)|^2$ are shown in Fig. 1b. The transmittance and

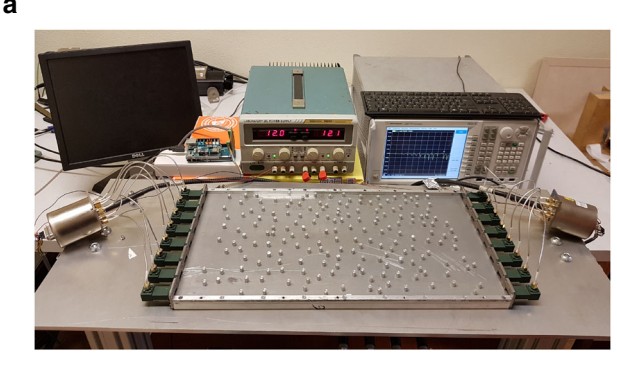

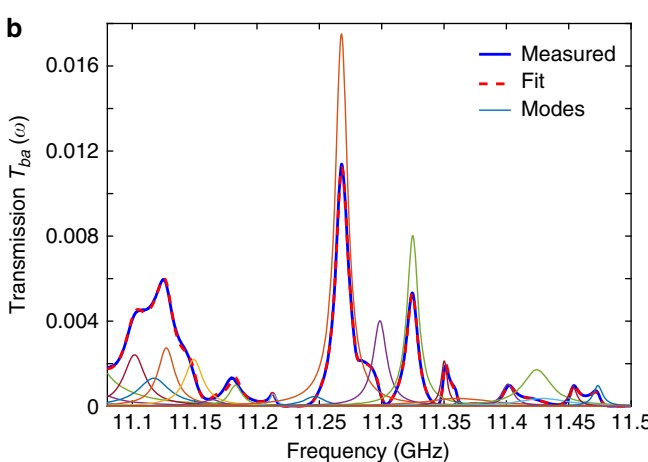

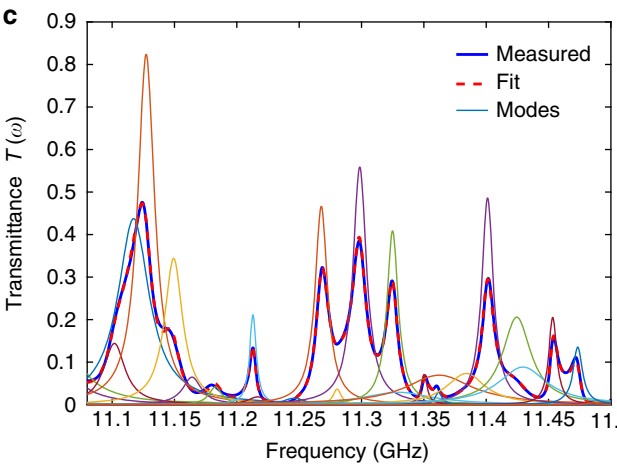

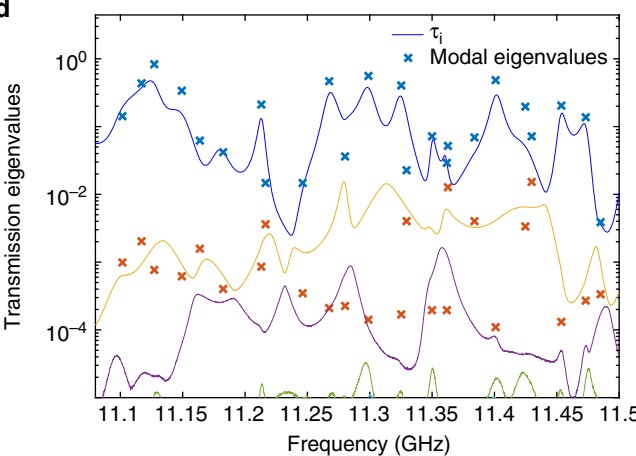

**Fig. 1** Experimental setup and decomposition of the TM into modes. **a** Experimental setup with the top plate removed to show the cavity with 300 randomly-positioned aluminum disks. The TM is measured between the 8 antennas on the left and right sides of the cavity. **b** Transmission (blue curve) between two channels, $|t_{ba}(\omega)|^2$, and its reconstruction found using HI (dashed red curve) in the [11–11.5] GHz range. The thin lines are modal strengths in transmission, $|t_{ab}^n|^2|\varphi_n(\omega)|^2$. **c** The measured transmittance, $T(\omega) = \Sigma_{ab}|t_{ba}(\omega)|^2$, (blue curve) and its reconstruction from the MTMs (dashed red curves), as well as the modal contributions $T_n(\omega)$ to the transmittance (thin lines). **d** The spectrum of transmission eigenvalues $\tau_i(\omega)$ and the first (blue crosses) and second (orange crosses) eigenvalues of the MTM on resonance are shown in a semilog plot. The second eigenvalue of the measured MTM is typically two orders of magnitude smaller than the first

the contribution $T_n(\omega) = \Sigma_{ba}|t_{ba}^n|^2|\varphi_n(\omega)|^2$ of each mode to $T(\omega)$ are shown in Fig. 1c. The reconstructions of the transmission and of the transmittance from the modes and the measurements of these quantities are in excellent agreement.

The degree of modal overlap may be expressed as the ratio of the mode width and spacing $\delta = \delta\omega/\Delta\omega$, where $\delta\omega = <\Gamma_n>$ is the average linewidth of modes and $\Delta\omega = <\omega_{m+1} - \omega_m>$ is the typical spacing between neighboring modes. In random media, the degree to which modes overlap spectrally tracks the spatial extent of the eigenstates in the interior of the sample and so the crossover from diffusion to localization[44,46,52,53]. When reflection at the interface is weak, the ensemble average of the ratio of the level width to level spacing gives the Thouless number, which is equal to the conductance $g$[44]. Modes of the medium are generally exponentially peaked within the sample when $\delta < 1$ and extended when $\delta > 1$. Here the average linewidth is $\langle\Gamma_n\rangle \sim 9$ MHz and the degree of modal overlap is $\delta \sim 1.2$. The high average modal quality factor $Q = 3300$ justifies the assumption that the coupling vectors between the quasi-normal modes and the antennas are independent of frequency.

An additional check is placed on the accuracy of the modal decomposition of the TM when the fit to transmission is carried out simultaneously at several points in the sample: the rank of MTMs found in the fits of transmission must be close to unit rank, as predicted by Eq. (4). This additional check is not possible when a single spectrum of a transmission coefficient is decomposed into modes with use of HI, as has been done in studies of chaotic cavities[45]. Spectra of the transmission eigenvalues of the measured TM, $\tau_i(\omega)$, are compared in Fig. 1d to the transmission eigenvalues of modes found from the diagonalization of $t_n t_n^\dagger$. The second modal eigenvalue is typically smaller than the first by a factor of $10^{-2}$. The dominance of the first modal eigenvalue supports the predicted decomposition of the TM into MTMs of unit rank. This is further confirmed in measurements with smaller modal overlap (Supplementary Note 1) in which the ratios of the second and first modal eigenvalue are substantially smaller. The ratio is still smaller in simulations for samples with modal overlap comparable to that in experiments indicating that the quality of the modal decomposition is degraded by noise in the measurements.

We find that when the first and second eigenvalues of the MTM are close in value, the results are likely to be spurious. The modal analysis is limited here to samples with moderate modal overlap. For higher modal overlap, spurious resonances may appear in the modal analysis due to the contributions of modes with large linewidth which cannot be resolved and to modes that lie outside the frequency range but still contribute since their linewidth is broad.

**Modal selectivity.** The strength of excitation of an individual mode is maximized when the incoming wave on the left or right excites the sample with the optimal modal patterns $\mathbf{W}_{Ln}^*$ and $\mathbf{W}_{Rn}^*$, respectively. These are the complex conjugates or the time-reversal of modal speckle patterns at the sample boundaries. Using Eq. (4), the vector of the transmitted field for an excitation of the sample from the left with the normalized optimal waveform, $\mathbf{v}_n = \mathbf{W}_{Ln}^*/\mathbf{W}_{Ln}$, can be expressed as

$$\mathbf{E}_{max}(\omega) = \mathbf{t}(\omega)\mathbf{v}_n = -i\frac{\mathbf{W}_{Ln}^2}{\omega - \omega_n + i\frac{\Gamma_n}{2}}\frac{\mathbf{W}_{Rn}}{\mathbf{W}_{Ln}}$$
$$-i\Sigma_{m\neq n}\frac{\mathbf{W}_{Lm}^T\mathbf{W}_{Ln}^*}{\left(\omega - \omega_m + i\frac{\Gamma_m}{2}\right)}\frac{\mathbf{W}_{Rm}}{\mathbf{W}_{Ln}}. \quad (5)$$

The first term in Eq. (5) gives the contribution of the $n^{th}$ mode to transmission for maximal coupling and the sum in the second term gives the contributions of other modes. Apart from the Lorentzian function, the energy in the mode to which the field is maximally coupled is equal to $\|\mathbf{W}_{Ln}\|^2\|\mathbf{W}_{Rn}\|^2$. This can be compared to the average energy for a normalized random incoming waveform $\mathbf{v}_{rand}$ which is $\left\langle|\mathbf{W}_{Ln}^T\mathbf{v}_{rand}|^2\right\rangle\|\mathbf{W}_{Ln}\|^2 = \|\mathbf{W}_{Ln}\|^2\|\mathbf{W}_{Rn}\|^2/N$. The energy in the mode for maximal coupling in an $N$-channel system is therefore enhanced by a factor $N$ using the optimal modal pattern. This property is a consequence of the unit-rank of the MTMs. At the same time, the contribution to transmission of the selected mode vanishes for any incoming vector orthogonal to the optimal modal pattern $\mathbf{W}_{Ln}$. Residual transmission is due to the contributions of neighboring modes.

Excitation of specific quasi-normal modes differs from excitation of transmission eigenchannels. The eigenchannels and eigenvalues of the TM can be found via a singular value decomposition in which the TM at a single frequency is expressed as the product of three $N \times N$ matrices, $\mathbf{t}(\omega) = \mathbf{U}\Lambda\mathbf{V}^\dagger$. Here $\Lambda$ is a diagonal matrix whose elements are the singular values $\sqrt{\tau_i}$, and $\mathbf{V}$ and $\mathbf{U}$ are unitary matrices and correspond to the waveforms of the transmission eigenchannel on the input and output of the sample, respectively. In contrast to modes, which have a Lorentzian spectrum, the eigenchannels are defined at a specific frequency; a new set of transmission eigenvalues and eigenchannels must be computed at each frequency[5,23,54,55]. However, the spectral characteristics of the channels can be obtained by decomposing the transmission eigenchannels into modes[23,54]. When a single mode dominates transmission, the MTM for this mode is close to the first eigenchannel. At the resonance, the first transmission eigenvalue is, $\tau_1(\omega_n) = \|\mathbf{W}_{Ln}\|^2\|\mathbf{W}_{Rn}\|^2/(\Gamma_n/2)^2$. When several resonances overlap, however, the first transmission eigenchannel is a combination of modal contributions of several modes.

**Experimental demonstration of modal selectivity.** To explore the degree of modal selectivity for different incident waveforms, the field coefficient within the medium $e_a(x, y, \omega)$ is measured using a wire antenna inserted through subwavelength holes (see Methods). The contributions of modes inside the medium, $e_a^n(x,y)$, are then obtained from a fit of the coefficients $e_a(x,y,\omega) = \Sigma_n e_a^n(x,y)\varphi_n(\omega)$ using the set of resonances $\{\omega_n, \Gamma_n\}$ obtained from the modal expansion of the TM. The spatial energy distribution for each mode is reconstructed from the contributions of modal field patterns due to the eight incoming channels summed to give the optimum incident modal pattern.

For isolated modes, strong modal discrimination is readily accomplished by tuning to resonance. The strength of excitation is enhanced over the average of random excitation by a factor of $N$ by adjusting the incident wavefront to the optimal modal pattern. In Fig. 2, we consider two weakly overlapping modes at $f_1 = = 11.434$ GHz and $f_2 = 11.461$ GHz with linewidths of $\Gamma_1/(2\pi) = 3.75$ MHz and $\Gamma_2/(2\pi) = 4.84$ MHz. This gives a degree of modal overlap between the modes of $\delta_{12} = \left[\frac{\Gamma_n + \Gamma_{n+1}}{2}\right]/(\omega_{n+1} - \omega_n)$ of 0.15. The two modes are spatially distinct and peaked at different points within the sample, as seen in Fig. 2a. For maximal coupling to the first and second modes, the transmission is seen in Fig. 2c, d to be enhanced by a factor of close to $N = 8$ at the resonance of the two modes in comparison transmission for a random incoming wavefront shown in Fig. 2b. The energy density inside the medium at resonance then closely matches the spatial distribution of the mode, as seen in the insets of Fig. 2c, d. In contrast, for vanishing coupling to the first mode using the third singular vector of the MTM, the first mode does not contribute to transmission and the energy density is due to the contribution of

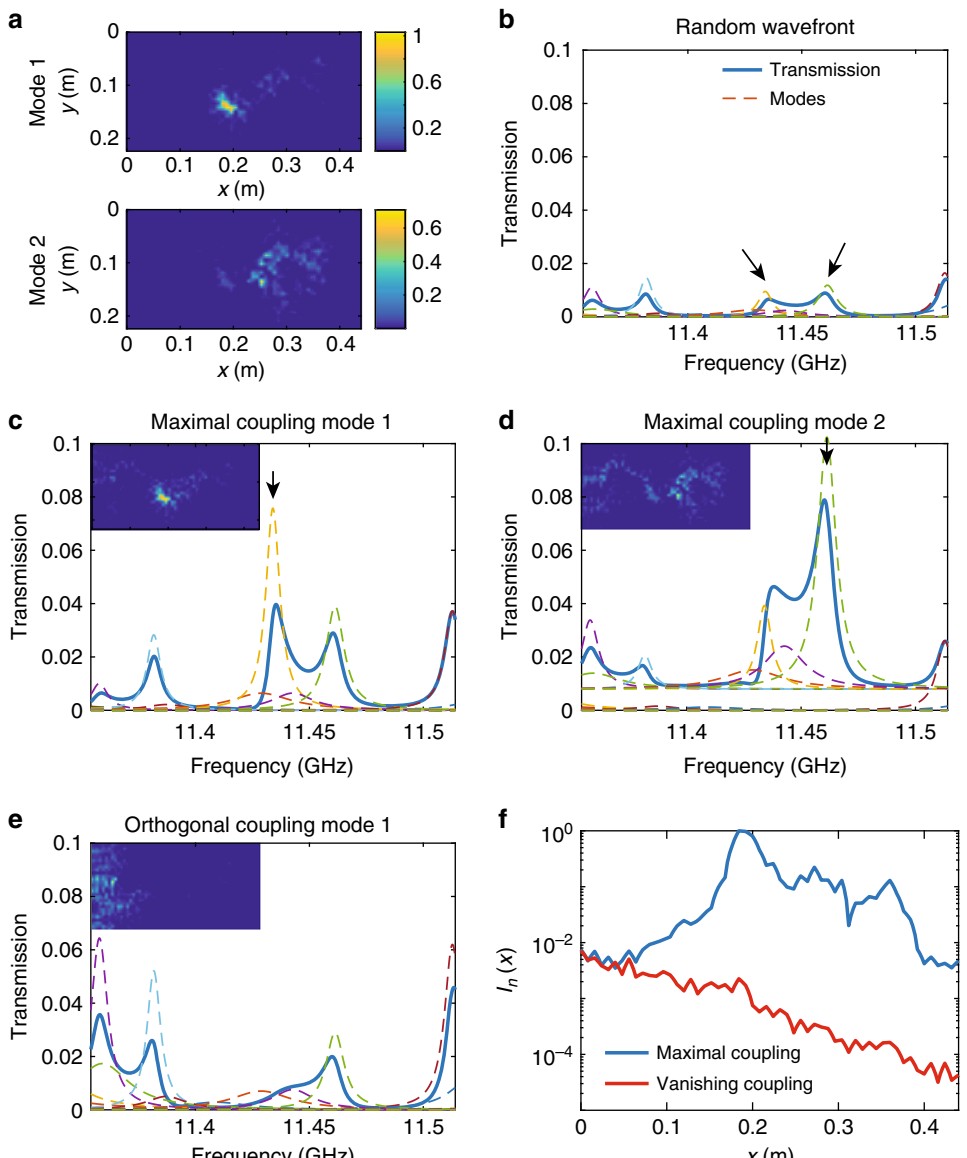

**Fig. 2** Selectivity in modal excitation of weakly overlapping modes. **a** Spatial profiles of the two modes whose resonant frequencies are $f_1 = 11.434$ GHz and $f_2 = 11.461$ GHz are indicated with arrows in **b** . **b–e** Transmission and strengths of modes in transmission (dashed lines) for **b** a random incoming wavefront, **c**, **d** maximal coupling to the first mode (**c**) and second mode (**d**) indicated with arrows. **e** Vanishing coupling to the first mode. The insets in **b–e** are spatial intensity profiles determined from measurements. **f** Modal energy density profiles inside the sample averaged over the cross-section for the first mode is shown for maximal coupling (blue curve) and vanishing coupling (red curve)

weakly overlapping modes, as seen in Fig. 2e. The energy density is concentrated at the beginning of the sample and falls rapidly into sample. Thus, specific modes can be selected using optimal modal incident wave patterns when the modes overlap weakly.

In Fig. 2f, we show the average over the cross-section of the modal strength inside the medium for maximal and vanishing coupling to the first mode at $f_1 = 11.434$ GHz. The contribution of the mode is maximum for the optimal incident modal pattern and should vanish for the orthogonal waveforms. For vanishing coupling, the intensity is seen to fall exponentially within the sample and transmission is more than two orders of magnitude below that for maximal coupling, in agreement with the ratio between the first and second modal eigenvalues.

We next consider selectivity in a case of two strongly overlapping modes (Fig. 3). The modes at $f_1 = 11.763$ GHz and $f_2 = 11.773$ GHz with linewidths $\Gamma_1/(2\pi) = 12.1$ MHz and $\Gamma_2/(2\pi)$

$= 16.6$ MHz have the modal overlap factor $\delta_{12} = 1.35$. The spatial profiles of the two modes are seen in Fig. 3a to be more extended than the modes discussed previously with $\delta_{12} = 0.15$ and to be very similar. In addition to enhancing the contribution of the maximally excited mode, maximal coupling is seen in Fig. 3c, d to enhance the contribution of the neighboring mode in comparison to a wavefront that has not been optimized. The modal transmission associated with the second mode is enhanced by a factor 4 for maximal coupling to the first mode.

The distributions of energy density for a random wavefront and for the first transmission eigenchannel at a frequency midway between the two resonances $\omega_0 = (\omega_n + \omega_{n+1})/2$, are seen in Fig. 4a, b to be primarily mixtures of the modal spatial patterns of the two neighboring modes shown in Fig. 3[54]. Nevertheless, it is possible to preferentially excite a single mode by adjusting the incident wave to match the pattern of one of the nearby resonant

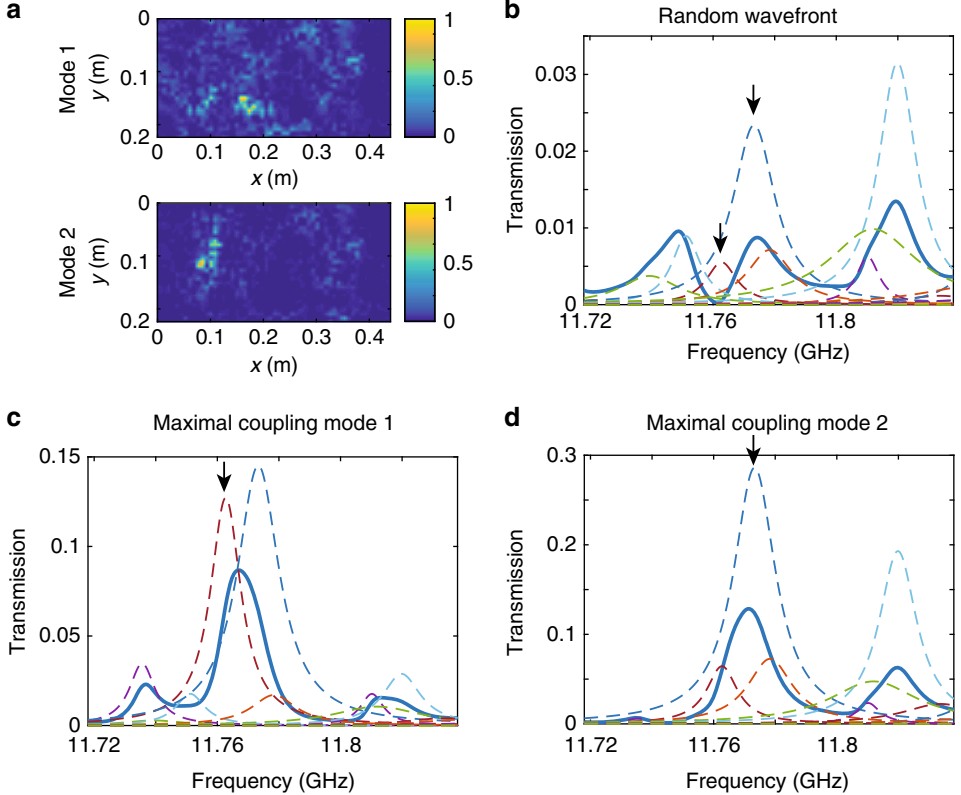

**Fig. 3** Selectivity in modal excitation of strongly overlapping modes. **a** Spatial profile of the energy density of two modes at at $f_1 = 11.763$ GHz and $f_2 = 11.773$ GHz. **b-d** Transmission (blue curve) and strengths of modes (dashed curves) for **b** a random incoming wavefront, and **c**, **d** for the wavefront which maximally couple to the modes peaked at the frequencies indicated with arrows

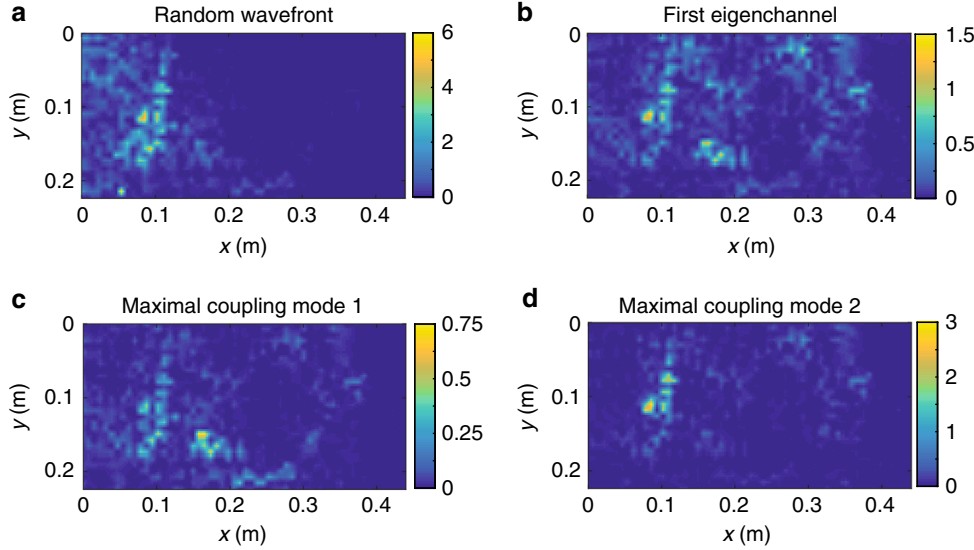

**Fig. 4** Selectivity between resonances of overlapping modes. **a-d** Spatial intensity distributions at frequency $(f_1 + f_2)/2$ for **a** a random incoming wavefront, **b** the first transmission eigenchannel, and **c**, **d** maximal coupling to the first (**c**) and second (**d**) modes

modes. In Fig. 4c, d, the energy density at the frequency between the modal resonances is shown for maximal coupling to one or the other of the modes. In each case, the energy density matches the spatial distribution of the selected mode shown in Fig. 3a. The degree of modal selectivity between two modes achieved by maximizing the input for one of the modes is reduced as a result of the hybridization and spectral broadening of the modes of the

closed system when the sample is coupled to its surroundings. We will see in the theoretical analysis and measurements below that the similarity in modal patterns in the interior of the sample seen in Fig. 3a is a consequence of the bi-orthogonality of the eigenfunctions and the correlation between them.

The degree of modal selectivity can be further enhanced at the expense of the net excitation of the mode by exciting with a

waveform that is orthogonal to the neighboring mode. In the case of two overlapping modes, this is achieved by illuminating the sample with an incident wavefront $a\mathbf{W}_{L1}^* + b\mathbf{W}_{L2}^*$ with coefficients $a$ and $b$ satisfying the condition, $a\mathbf{W}_{L2}^T\mathbf{W}_{L1}^* + b\mathbf{W}_{L2}^T\mathbf{W}_{L2}^* = 0$. The contribution of the second mode therefore vanishes so that the desired modes is perfectly selected relative to the second mode. Because of the correlation of modal speckle pattern, this also suppresses the excitation of the selected mode.

### Mixing of eigenfunctions

**Nonorthogonality matrix**. Equation (4) shows that the contribution to the output speckle pattern of the $m$th mode for the incident waveform that couples maximally to the $n$th mode at resonance, $\omega = \omega_n$, is equal to

$$C_{mn} = \frac{\mathbf{W}_{Lm}^\dagger \mathbf{W}_{Ln}}{\omega_n - \tilde{\omega}_m^*}. \qquad (6)$$

The matrix $C$ involves the degree of correlation between the modal patterns at the input. It can be related to the Bell-Steinberger nonorthogonality matrix $\mathbf{U}$, which gives the correlation over the volume between the eigenfunctions $\varphi_n$ of the system[32,56]. The elements of $\mathbf{U}$ given by the scalar product, $U_{mn} = \phi_m^\dagger \phi_n$ can also be expressed in terms of the vectors $\mathbf{W}_n$, in the absence of losses that are not due to the coupling of the antennas to the system.

$$U_{mn} \equiv \phi_m^\dagger \phi_n = i \frac{\mathbf{W}_m^\dagger \mathbf{W}_n}{\tilde{\omega}_n - \tilde{\omega}_m^*} \qquad (7)$$

Here, the vector $\mathbf{W}_n$ with $2N$ elements is the concatenation of the two vectors $\mathbf{W}_{Ln}$ and $\mathbf{W}_{Rn}$. In Hermitian systems, $U$ is the identity matrix since the modes are orthogonal over the volume. However for non-Hermitian systems, the diagonal elements of $\mathbf{U}$, which are the inverses of the phase rigidities of the eigenfunctions, $U_{nn} = 1/\rho_n$, increase with modal overlap[32,57,58]. The phase rigidity, $\rho_n \equiv \langle \phi_n^2 \rangle / \langle |\phi_n| \rangle^2$, can be expressed in terms of the degree of complexness of the eigenfunctions of $H_{\text{eff}}$, $q_n^2 = \langle \text{Im}(\phi_n)^2 \rangle / \langle \text{Re}(\phi_n)^2 \rangle$, as $\rho_n = (1 - q_n^2)/(1 + q_n^2)$[39]. For traveling waves, the real and imaginary parts of the eigenstates are the same on average so that $q_n = 1$ and $\rho_n = 0$. In contrast, for isolated resonances, the eigenfunctions coincide with the real eigenfunctions of the closed system and $\rho_n \to 1$. In random media, the degree of complexness and the phase rigidity track the transition from the diffusive to localized regime[59].

The completeness of the eigenfunctions implies the sum rule $\Sigma_m U_{nm}^2 = 1$[60]. The positive diagonal elements of $U$ matrix and the negative off-diagonal elements are enhanced as the degree of modal overlap $\delta$ increases. In the weak coupling regime, the diagonal elements are of order $1 + \delta^2$[48], while the magnitude of off-diagonal elements increase as $-\delta$[58].

Equation (7) shows that the non-orthogonality of eigenfunctions over the volume yields a non-vanishing degree of correlation between modal speckle patterns at the interface. In the case of maximal coupling to a mode, the off-diagonal elements are also responsible of the non-vanishing of the contribution of the neighboring modes as shown by the similarity of Eqs. (6) and (7). However, two differences can be observed. First, $C_{nm}$ involves $\mathbf{W}_{Lm}^\dagger \mathbf{W}_{Ln}$ instead of $\mathbf{W}_m^\dagger \mathbf{W}_n = \mathbf{W}_{Lm}^\dagger \mathbf{W}_{Ln} + \mathbf{W}_{Rm}^\dagger \mathbf{W}_{Rn}$. Since $\mathbf{W}_{Ln}$ and $\mathbf{W}_{Rn}$ are statistically independent random variables, we may apply the central limit theorem and approximate $\mathbf{W}_m^\dagger \mathbf{W}_n \sim 2\mathbf{W}_{Lm}^\dagger \mathbf{W}_{Ln}$ for $N \gg 1$.

Second, the denominator of Eq. (6) depends on $\omega_n - \tilde{\omega}_m^*$ instead of $\tilde{\omega}_n - \tilde{\omega}_m^*$. However, in the case of strongly overlapping resonances, the spacing between the central frequencies $\omega_n - \omega_m$ is much smaller than the linewidths $\Gamma_n$ and $\Gamma_m$ so that $\omega_n - \tilde{\omega}_m^* \sim -i\Gamma_m/2$ and $\tilde{\omega}_n - \tilde{\omega}_m^* \sim -i(\Gamma_n + \Gamma_m)/2$. For resonances with similar linewidths, we therefore obtain $C_{nm} \sim U_{nm}$. The non-orthogonality of eigenfunctions therefore yields a non-vanishing degree of correlation between modal speckle patterns.

We define the modal selectivity for maximal modal coupling as the ratio of the strength of the selected mode in transmission over the incoherent sum of strengths of all modes

$$S_{\text{mode}} = \frac{T_n(\omega_n)}{\Sigma_{m=1}^M T_m(\omega_n)} \sim \frac{|C_{nn}|^2}{\Sigma_m |C_{nm}|^2}. \qquad (8)$$

A diagonal matrix $C$ would correspond to perfect modal selectivity with $S_{\text{mode}} = 1$. However, for non-Hermitian systems, modal selectivity falls below unity due to off-diagonal elements of $C_{nm}$. Thus the bi-orthogonality of the eigenfunctions of a non-Hermitian system reduces the degree of modal selectivity for maximal coupling to a mode. This is illustrated in the analytical analysis of a two-level non-Hermitian effective Hamiltonian model.

**Two-level effective Hamiltonian**. The modes of the system are expressed in the basis of the two modes of the closed cavity[39,40]

$$\mathbf{H}_{\text{eff}} = \begin{pmatrix} \omega_1 & 0 \\ 0 & \omega_2 \end{pmatrix} - \frac{i}{2} \begin{pmatrix} \Gamma_{11} & \Gamma_{12} \\ \Gamma_{12} & \Gamma_{22} \end{pmatrix} \qquad (9)$$

The parameters $\Gamma_{nm}$ are given by, $\Gamma_{nm} = \Sigma_{c=1}^{2N} V_n^c V_m^c$, where the vectors $V_n$ represent the coupling of the closed system to the leads (see Eq. (1))[39]. The parameter $\gamma = \Gamma_{12}$ is the coupling parameter between the resonances. Diagonalizing $\mathbf{H}_{\text{eff}}$ gives the eigenvalues $\tilde{\omega}_{1,2} = \frac{1}{2}(\omega_2 + \omega_1) \mp \frac{1}{2}\sqrt{\epsilon^2 - \gamma^2} - \frac{i}{4}(\Gamma_{11} + \Gamma_{22})$, with $\epsilon = (\omega_2 - \omega_1) - \frac{i}{2}(\Gamma_{11} - \Gamma_{22})$. The eigenvectors $|\phi_n\rangle$ of the effective Hamiltonian $\mathbf{H}_{\text{eff}}$ can be written in the basis $\{|\psi_n\rangle\}$ of the unperturbed eigenvectors of the Hamiltonian of the closed system $H$[48]

$$|\phi_1\rangle = \frac{1}{\sqrt{1 - f^2}} \begin{pmatrix} 1 \\ -if \end{pmatrix}, \quad |\phi_2\rangle = \frac{1}{\sqrt{1 - f^2}} \begin{pmatrix} if \\ 1 \end{pmatrix}. \qquad (10)$$

The mixing of the two eigenstates depends on the single parameter, $f = \gamma/(\epsilon + \sqrt{\epsilon^2 - \gamma^2})$. The degree of complexness of the eigenfunctions is the same for both modes, $q_1^2 = q_2^2 = f^2$. In the present case of the two-level Hamiltonian, $q_n^2$ increases from 0 for isolated modes ($\gamma \ll \epsilon$) to unity for $f = 1$, which is the case of an exceptional point at which the eigenvalues $\tilde{\omega}_1$ and $\tilde{\omega}_2$ coalesce[61].

The two-level Hamiltonian model is illustrated in Fig. 5 by the eigenfunctions of two hybridized modes at $f_1 = 11.260$ GHz and $f_2 = 11.266$ GHz, which are isolated from other modes but overlap strongly with a degree of overlap between the modes of $\delta_{12} = 5.5$. The eigenfunctions are normalized following the bi-orthogonality condition, $\int^d r \phi_n^2(r) = 1$. There is a strong similarity between $\text{Re}(\phi_1)$ and $\text{Im}(\phi_2)$ and between $\text{Im}(\phi_1)$ and $\text{Re}(\phi_2)$, as anticipated in Eq. (10). The eigenfunctions give $q_1^2 = 0.86$ and $q_2^2 = 0.4$. The two values are not equal because of the weak overlap with other modes. We observe in Fig. 5e, f that when the incident wave is maximally coupled to the first or second mode, the transmission spectra and the contribution of

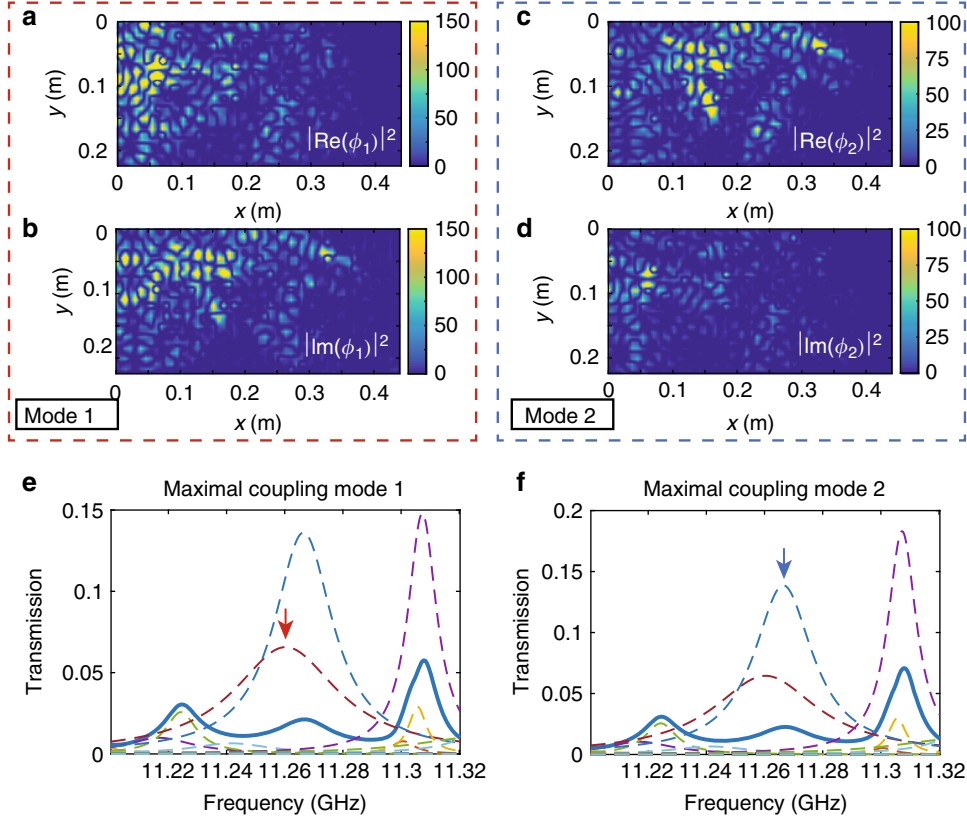

**Fig. 5** Modal mixing. **a–d** The square of the real and imaginary parts, $\mathrm{Re}(\phi_n)^2$ (**a**, **c**) and $\mathrm{Im}(\phi_n)^2$ (**b**, **d**), of two strongly overlapping modes with resonance at $f_1 = 11.260$ GHz and $f_2 = 11.266$ GHz with linewidths $\Gamma_1/(2\pi) = 21.7$ MHz and $\Gamma_2/(2\pi) = 12.3$ MHz, respectively. **e**, **f** Transmission spectra (blue line) for maximal coupling to the first (**e**) and the second (**f**) mode. The dashed lines are the modal strengths. The small value of transmission in comparison to modal strengths in **e**, **f** is a result of strong destructive interference between the highly correlated speckle patterns of strongly overlapping modes

the two modes are very similar. The correlation between the incident waveforms $W_{L1}$ and $\mathbf{W}_{L2}$, $\left|\mathbf{W}_{L1}^{\dagger}\mathbf{W}_{L2}\right|/(\|\mathbf{W}_{L1}\|\|\mathbf{W}_{L2}\|)$ is 0.98. The modal mixing of two strongly overlapping modes and its impact on the modal selectivity are further confirmed in finite-element simulations in Supplementary Note 2.

The decrease of $S_{\mathrm{mode}}$ with increasing degree of modal mixing is demonstrated analytically in Supplementary Note 3 within the framework of the two-level Hamiltonian model. By expressing the average degree of correlation between vectors $\mathbf{W}_{L1}$ and $\mathbf{W}_{L2}$ as a function of $f$ in the limit $N \gg 1$ and $f \ll 1$, we find using Eq. (4) that $S_{\mathrm{mode}}$ only depends upon the modal overlap and $f$ with $S_{\mathrm{mode}} \sim \left[1 + \frac{4f^2\Gamma_1^2}{4\Delta_{12}^2 + \Gamma_2^2}\right]^{-1}$, where $\Delta_{12} = \omega_2 - \omega_1$.

**Average modal selectivity**. In order to investigate the average modal selectivity in a large number of samples, we carry out simulations utilizing the recursive Green's function method[62] in random quasi-1D samples connected to leads supporting $N$ channels to find $t(\omega)$ (see Methods). Four ensembles with modal overlap $\delta$ equal to 0.08, 0.11, 0.64, and 1.13 are studied. The number of channels is $N = 10$ for $\delta = 0.08$, $N = 16$ for $\delta = 0.11$ and 0.64, and $N = 33$ for diffusive samples with $\delta > 1$. The HI method is applied to more than one hundred samples for each ensemble giving more than 4,000 modes.

The modal selectivity for maximal coupling $S_{\mathrm{mode}}$ is shown as a function of the modal overlap $\delta$ in Fig. 6, and compared to $S_{\mathrm{ran}}$ computed for a random incident wavefront. As expected, maximal coupling enhances modal selectivity, but $S_{\mathrm{mode}}$ falls below the value expected in the case of isolated modes of unity,

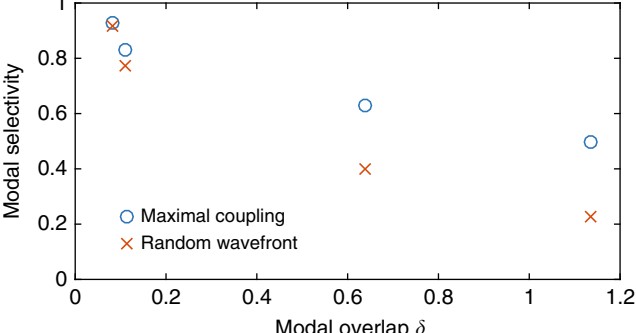

**Fig. 6** Average modal selectivity. The average modal selectivity for maximal coupling $S_{\mathrm{mode}}$ (blue circles) and for a random wavefront $S_{\mathrm{ran}}$ (red crosses) is plotted for four ensembles as a function of the modal overlap $\delta$

even for localized waves. For $\delta = 0.08$, modes overlap and interfere giving $S_{\mathrm{mode}} = 0.92$. Modal selectivity is seen to decrease with increasing $\delta$ as a result of greater spectral overlap with a larger number of modes and consequent increased correlation between their MTMs.

## Discussion

We have considered the inverse problem of characterizing and controlling the modes within the sample on the basis of the properties of waves scattered from the sample. We have seen that the road to the control of modes and of wave properties related to modes runs through the MTM which can be obtained from

spectra of the TM. We have demonstrated that a single mode can be excited within the sample even in the case of a moderate modal overlap. The incident vector of the MTM of unit rank couples maximally to the mode with an enhancement by a factor $N$ over excitation by a random wavefront. However, as modal overlap increases as a result of greater coupling of the modes to the environment through the boundaries of the sample, the bi-orthogonality of the eigenfunctions leads to increasing modal correlation so that the degree of modal control is reduced.

This investigation of selective excitation of quasi-normal modes in disordered system has been carried out in systems with resonances with high quality factors. The decomposition of the TM into MTMs is more challenging for diffusive waves when the degree of modal overlap is high. However, as Alpeggiani et al. have demonstrated analytically, the scattering matrix can be reconstructed from the far-field properties of the eigenmodes for any degree of overlap[26]. The modal coefficients depend on all other contributing modes through a coupling matrix, but MTMs are still of unit rank. Future studies will be dedicated to exploring the characteristics of modes and the limits of selectivity in systems with strong modal overlap. This will advance a more comprehensive understanding of the relationship between eigenchannels, time-delay eigenstates and quasi-normal modes of open system.

Selecting specific modes in samples in which the wave is localized would make it possible to deliver energy to specific regions of a sample. In the case of moderate modal overlap in open random systems, the modes extend over the entire sample, even in absorbing samples, and so if a single mode or a small number of modes is selected, it would be possible to deliver energy to the center of the sample. In contrast, when, many modes overlap, the average profile of energy density within the sample is determined by the diffusion equation and is concentrated within an absorption depth of the sample $L_a = (D\tau_a)^{1/2}$, where $D$ is the diffusion coefficient and $\tau_a$ is the absorption time, and particularly the absorption associated with exciting gain in the medium[63]. As a result, light emitted from excited samples with weakly overlapping modes, in which energy penetrates more deeply into a random medium, is longer lived than in samples in with stronger modal overlap. There is therefore greater opportunity for the emitted photons to stimulate emission before escaping the sample. The lasing threshold will consequently be lowered and narrow-line emission will be observed[64]. In general the smaller the degree of modal overlap, the more it is possible to deposit energy into a random absorbing medium. Finding the MTM and then pumping from the front of the sample[64,65] could thereby lower the threshold of random lasers via coherent feedback.

Modal decomposition of the TM has been demonstrated with microwave radiation but is in principle possible in optics. Measurement of spectrally resolved TM has indeed been reported recently[66]. Modal selectivity could then be utilized to enhance light-matter interactions in photonic materials[67], solar cells[68], or biomedical optics[69]. The use of MTMs is not restricted to random media but can also be applied to structured media such as optical microcavities[31], or photonic crystals[70].

and receiving antennas are used. The spacing between two antennas on the left and right sides of the sample is metallic as seen in Fig. 1a.

The field inside the waveguide is detected by inserting an antenna sequentially into a square grid of holes which are 4 mm in diameter and spaced by 8 mm on a side. The transmission coefficient $e_a(x, y, \omega)$ is measured between each source antenna and a wire antenna inserted 0.5 mm below the bottom of the 6-mm thick aluminum cover. The penetration depth of the antenna is small enough that it does not distort the field profile in the waveguide. The spatial energy distribution $I(x, y)$ for any incoming vector $v$ is then reconstructed from the coherent superposition of the fields arising from each source antenna, $I(x, y) = |\Sigma_a e_a(x, y)v_a|^2$.

Once the decomposition into modes of the field inside the sample, $e_a(x, y, \omega) = \Sigma_n e_a^n(x, y)\varphi_n(\omega)$, has been obtained, the modal spatial profile is found from an average on incoming channels $I_n(x, y) = \langle |e_n^n(x, y)|^2 \rangle$. We also present in Fig. 2f the modal energy density profile for incoming vector $v$ computed from $W_n(x, y) = |\Sigma_a e_a^n(x, y)v_a|^2$. For maximal modal coupling $v = W_{Ln}/||W_{Ln}||$ and for vanishing coupling $v$ is orthogonal to $W_{Ln}$.

**Harmonic inversion**. The modal analysis of the TM is performed using the HI method to obtain the complex modal frequencies $\omega_m - i\Gamma_m$ within a range $\omega_{min} < \omega_m < \omega_{max}$. Following the algorithm described in ref. [50], we choose a Fourier-type Krylov basis to extract the resonances. The HI method consists of solving a generalized eigenvalue problem applied to two matrices $U^{(0)}$ and $U^{(1)}$ of dimension $J \times J$ with $J = N' dt(\omega_{max} - \omega_{min})/(4\pi)$ which are created from a time signal of length $N'$ and time step $dt$. The generalized eigenvalue problem is $U^{(1)}B_n = u_n U^{(0)}B_n$, where $u_n$ and $B_n$ are the eigenvalues and the eigenvectors, respectively. The $M$ non-zero eigenvalues yield the $M$ unknown complex frequencies; the associated complex amplitudes can be computed from the eigenvectors.

However, we may miss a few modes by applying HI to a single spectrum of the TM since the modal speckle patterns are random vectors. The strength of a particular mode could thus vanish for a single spectrum when the incoming and outgoing channels correspond to two nodes of this mode. We therefore include several spectra from the TM in the generalized eigenvalue problem. We first perform an inverse Fourier transform of nine field transmission spectra randomly chosen from the $N^2$ spectra of the TM. The time window of the inverse Fourier transform is taken to be the time domain for which the signals are above the noise level. We then create 9 matrices $U_i^{(0)}$ and $U_i^{(1)}$ from each time signals $s_i(t)$. Those 9 matrices of dimensions $J \times J$ are then concatenated into two matrices of dimension $3J \times 3J$, $U_T^{(0)}$ and $U_T^{(1)}$. Because we are seeking for the same set of resonance time signals $s_i(t)$, we solve the generalized eigenvalue problem $U_T^{(1)}B_n = u_n U_T^{(0)}B_n$. We then extract the resonances $\tilde{\omega}_n$ from the significant eigenvalues $u_n$. However, this does not directly provide the associated modal transmission coefficients $t_{ba}^n$ which are the elements of the MTMs. These are then obtained from a simple inverse problem by fitting in the time domain each transmission coefficient of the TM using Eq. (4) of the main text: $t_{ba}(\omega) = \Sigma_n \frac{t_{ba}^n}{\omega - \omega_n + i\Gamma_n/2}$.

**Recursive Green's function simulations**. The Green's functions between points at the input and output surfaces of a waveguide are obtained by solving the two-dimensional wave equation $\nabla^2\psi(x, y, \omega) + k_0^2\epsilon(x, y)\psi(x, y, \omega) = 0$ using the recursive Green's function method. The random dielectric permittivity $\epsilon(x, y)$ is drawn from a rectangular distribution centered on unity. The field transmission coefficients between each of the incoming modes $a$ and outgoing modes $b$ at frequency $\omega$, $t_{ba}(\omega)$, are then calculated using the projection of the Green's function onto the modes of the empty waveguide. Spectra of the TM is then obtained by computing the TM for each frequency over a frequency range which is much larger than the typical linewidth of the resonances.

## Data availability

The authors declare that all data that support the findings of this study are available from Matthieu Davy at matthieu.davy@univ-rennes1.fr upon reasonable request.

## Methods

**Experimental setup**. The aluminum cavity has length $L = 500$ mm, width $W = 268$ mm, and height $H = 8$ mm and only supports a single waveguide mode in the vertical dimension over the frequency range of the measurements. The antennas are waveguide to coax adapters designed for the Ku band (12–18 GHz). Measurements of individual elements of the TM between two antenna arrays are made with use of electro-mechanical switches and a vector network analyzer. The channels of the switches that are turned off are matched to a 50 Ω load so that the boundary conditions of the system do not change when different emitting source

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

## Acknowledgements

This publication was supported by the European Union through the European Regional Development Fund (ERDF), by the French region of Brittany and Rennes Métropole through the CPER Project SOPHIE/STIC & Ondes, and by the National Science Foundation under grant DMR/-BSF: 1609218. We would also like to acknowledge Cécile Leconte for her help in automating the scan and Zhou Shi, Yan Fyodorov, Ulrich Kühl, Olivier Legrand, Fabrice Mortessagne, Jing Wang, and Eli Ashoush for stimulating discussions.

## Author contributions

M.D. carried out the experiment, numerical simulations, and theoretical analysis. A.Z.G. and M.D. conceived the project, discussed the results, and wrote the manuscript.

## Additional information

**Competing interests:** The authors declare no competing interests.

