## [Peer Review File · Nature Communications]

Reviewers' Comments:

Reviewer #1:

Remarks to the Author:

The authors present a study of the modal contributions to the transmission matrix (TM) in a scattering medium. Overall the work is of high quality and addresses a very topical and timely question related to the decomposition of the TM in terms of modes. Upon reading the work, the reviewer formulated a lot of questions only to find them answered at a later point in the text. So altogether there are not too many points to address here.

In the context of the field of electromagnetic random media I believe the work represents a cutting edge contribution which deserves publication in Nature Communications.

Therefore I am happy to recommend acceptance of the work after they implement the following comments:

1. Overall the work is rigorous to the point of being quite technical. While many readers may be familiar with the framework of transmission matrices, perhaps more effort could be made in introducing the concept of modal transmission matrix, which is introduced on Page 2 as "which are the contributions of the modes to the TM [29]".

In hindsight, after reading the rest of the paper, it is clear what is meant. However already at this early point in the text the authors could be more clear, namely that the MTM is the decomposition of the TM in terms of individual modes of the scattering medium. So in simple terms, every mode in the system has an MTM and together they add up to the total TM.

2. Following up on point 1, perhaps it is not obvious for the reader what is the implication of the decomposition of the TM into the MTMs when going to eigenvalues. It is clear we can do singular value decomposition of the TM, however is the highest eigenvalue of the TM then related to the highest eigenvalue of each of the MTMs? This is not a priori clear and can be explained better. In other words, what happens with equation (3) when doing a singular value decomposition?

3. As a minor comment, there are two Equations (4). Also I found the notation of the sum in Eq. (3) somewhat confusing as it looks like a variable Σ_n among all the other greek variables with subscript n , but probably this is a typesetting issue.

4. Is there any limitation or assumption on the dimensionality of the system? It appears the theory is independent of dimensionality, the experiments are in 2D waveguide geometry. What about a slab geometry or open 3D system? For example, in a recent paper on arXiv: 1806.01917, it was found that eigenmodes are spatially localized in a slab. Does this affect the expansion of the TM into MTM?

5. In Equation (4) (the first one) a field E_{opt} is defined in terms of the transmission matrix and a normalized waveform. It was not entirely clear what was done here. Is E_{opt} supposed to be the optimized field after an optimization of the input state to maximize the total transmission? It seems W_n is the input pattern for only one mode, so it seems that E_{opt} only takes into account the input coupled into one mode n . But then it uses the full transmission matrix so it seems to measure the coupling of an input optimized for mode n , to all other modes. Perhaps this could be explained better in the text, we can work this out from the mathematics but it will help to get some intuition beforehand.

6. In Figure 1(d) they should explain better what are the different colored curves. One of the problems is that the different colors have not been explained in (b) and (c) either, so it is not clear what they represent.

7. In Figures 3 and 4 it appears that the two modes have a very similar pattern over a large part of their volume, while the difference seems to be mainly in the intensity within this pattern. At the point where this was discussed I had a lot of questions about this, which only became more clear when digesting the following theory on Page 10. I think it will help the reader if this similarity is already pointed out in the experimental results, where then can be referred to the theory further on explaining this correlation in more detail.

Reviewer #2:

Remarks to the Author:

Review of the manuscript entitled "Selectively exciting quasi-normal modes in open disordered systems" by Matthieu Davy and Azriel Z. Genack.

This work is concerned with a very fundamental aspect of wave transport in disordered systems, that is the relation between the transmission through the system, which results from the interaction of the wave with it, and its natural resonant modes - often called quasi-normal modes - which are intrinsic to the system and do not depend on the excitation. There has been a very intense activity in the past years on the use of wavefront shaping techniques to couple to individual "transmission eigenchannels". Here, the authors pose the question of how well a specific quasi-normal mode of the system can be excited, knowing the transmission matrix.

The two main messages of the work are the following: (i) via a modal decomposition of the transmission matrix, it is possible to enhance the energy coupled to a specific mode by a factor equal to the number of channels in the system; (ii) the modal selectivity is limited by the correlation between the wavefunctions of different modes and their spectral overlap. Theoretical arguments are given on both points. Point (i) is supported by experiments, while point (ii) is supported by numerics.

The methods appear to be correct and the results are scientifically sound. I believe that this work will interest the community of waves in random media and beyond, and should eventually be published. For reasons that will be detailed below, however, I am still uncertain about recommending publication of this work in Nature Communications. In brief, some conclusions of the work could benefit from a more thorough description and/or stronger experimental or numerical evidence; the generality and originality of the findings should be clarified (the theoretical formalism has limitations; some results are already expected from previous theories); the readability of the text and the quality of the figures could be improved.

I would like to leave open the possibility of recommending publication of this manuscript in Nature Communications and kindly ask the authors to address the points below.

IMPORTANT POINTS:

1/ The key mechanism from a microscopic point of view is the excitation of quasi-normal modes by an incident wavefield. It is known that the excitation coefficient of a quasi-normal mode at a given frequency can be expressed as an overlap integral of the incident field with the mode field. See for instance [P. Lalanne et al., *Laser & Photonics Reviews* 12, 1700113 (2018)] for a recent review on the topic. Considering that resonances of open systems are spectrally broad and their eigenfunctions non-orthogonal, it is then naturally expected that any incident field at a given frequency will also couple to other resonant modes. Supposing that the incident field is very close to that of the selected mode, the excitation to the other modes should be approximately given by the overlap between these modes and the selected one.

Such a concise and simple description of the problem at hand is currently lacking, or is not sufficiently emphasized, giving the wrong impression that the findings of the work should be

completely unexpected from the current literature.

Along this line, it is important also to remark that the physics into play here is basically that of coupled resonances, which have been studied quite exhaustively (e.g., for waveguide modes [W. P. Huang, *JOSA A* 11, 963 (1994)], or more recently for optical resonators [B. Vial and Y. Hao, *J. Opt.* 18, 115004 (2016)]). The interest in studying disordered systems from the application viewpoint is clear but when it comes to describing the physics of coupled systems, this knowledge base should be acknowledged.

I believe that, in the introduction, the authors should better explain what is expected from existing literature and clarify the novelty of their work (since the authors have, in fact, gone beyond this).

2/ Following the discussion above, the excitation in the system considered in this work cannot be completely arbitrary; it is driven by a finite number of sources in a system containing a finite number (N) of channels, thereby leading to a lower bound on the spatial field variation that could be achieved for the exciting field. The increase of the mode energy by a factor N compared to the random excitation is a very nice result, but:

- the origin of the factor N is not sufficiently argued and described, in my opinion. It is currently limited to 3 lines (from 98 to 100).
- the experimental proof is limited to a pair of modes (Fig. 2), while one would expect a larger statistics.

The authors should explain in detail where the factor N comes from in their theory and provide additional experimental results on mode energy enhancement with better statistics to be more convincing.

3/ The title and abstract suggest that the authors provide a general approach to the selective excitation of quasi-normal modes in open disordered systems. The theory that is developed here is however approximate, in the sense that leakage to the continuum is treated as a perturbation on a closed system. It is clear that the theoretical treatment will progressively fail for systems with lower quality factor resonators.

I believe that the authors should be extremely clear on this aspect in the manuscript. The generalization of their results to arbitrary systems could be discussed in the Discussion section.

4/ In the work, the authors support their various, successive claims by (i) experiments, (ii) finite-element simulations and (iii) recursive Green's function simulations on different system. It is not clear why the authors have chosen not to demonstrate all claims experimentally. The oscillation between various theories, an experiment and different numerical methods complicates the understanding of the work, especially when the choices made are not explained.

MINOR POINTS:

5/ Details on the experiment are given in the main text - not in the Methods - and the numerical analysis and numerical methods are in general not sufficiently described. They deserve sections in the Methods. In particular, more details about the reconstruction of the spatial energy distribution (lines 180-182) and the COMSOL simulations (which boundary conditions? which solver?) are needed.

6/ The figures are not excellent. There are no scales on the field maps, no description of the panels ("random wavefront", etc) in Fig. 3, no legend on Fig. 5, ... The frequencies of the modes in Fig. 2 do not seem to be 11.434 GHz and 11.461 GHz as stated in the text. Adding the variables (t_{ba} , δ , ...) on the figure axis titles would help.

7/ Some text is missing on line 302.

Response to comments of the reviewers

We thank both the reviewers for so carefully reviewing the manuscript and for so many perceptive and helpful comments. We have either implemented the suggestions in the revised manuscript or modified the manuscript to more clearly highlight the point at issue.

As a result, of the comments of the reviewers, the manuscript has been substantially revised and, we believe, significantly improved. Wherever possible, we have presented only the experimental results and eliminated the corresponding demonstration with simulations. We have also given a more expansive description of the background of this work. This has allowed us to more clearly highlight the most novel aspect of the paper, which is the characterization of correlation of modal speckle patterns in non-Hermitian systems and its impact upon the ability to select specific modes. The changes made are discussed in our response to each of the comments of the reviewers and are highlighted in yellow in the response below.

Reviewer #1

1. Overall the work is rigorous to the point of being quite technical. While many readers may be familiar with the framework of transmission matrices, perhaps more effort could be made in introducing the concept of modal transmission matrix, which is introduced on Page 2 as "which are the contributions of the modes to the TM [29]".

-- We have endeavored to make the article less technical in the sense that the reader is not assumed to be an expert in the field. Basic concepts are explained more fully and the essential physics is demonstrated in different ways only when this serves to bring new light to the subject. Thus we no longer show COMSOL simulations in cases when measurements are available.

We now provide a more expansive discussion of the modal transmission matrix (MTM) when it is first introduced. We write:

Here we consider the degree of modal selectivity that can be achieved in random media by manipulating the incident waveform. We approach the problem by analyzing the spectrum of the TM into its modal components in locally 2D N -channel samples. The complex modal frequencies and amplitudes are found by decomposing the spectra of the elements of the TM into a superposition of spectral lines via Breit-Wigner theory [29, 44, 45]

$$t_{ba}(\omega) = \sum_n \frac{t_{ba}^n}{\omega - \omega_n + i\Gamma_n/2} = \sum_n t_{ba}^n \varphi_n(\omega) . \quad (1)$$

Here $\varphi_n(\omega) = (\omega - \omega_n + i\Gamma_n/2)^{-1}$ is the frequency variation of excitation of the field associated with the mode with central frequency ω_n and linewidth Γ_n , and t_{ba}^n is the complex field transmission coefficient associated with the n^{th} resonance. Each resonance is then associated with a modal transmission matrix (MTM), t_n , which is built upon the coefficients t_{ba}^n , and is the contribution of a mode of the scattering medium to the TM [22]. An MTM therefore provides the incoming wavefront that maximally enhances the energy in a specific mode. However, modal selectivity becomes more challenging as the degree of modal overlap increases in non-Hermitian media.

In hindsight, after reading the rest of the paper, it is clear what is meant. However already at this early point in the text the authors could be more clear, namely that the MTM is the decomposition of the TM in terms of individual modes of the scattering medium. So in simple terms, every mode in the system has an MTM and together they add up to the total TM.

-- *This is now done in the highlighted paragraph above starting with “Here we consider.”*

2. Following up on point 1, perhaps it is not obvious for the reader what is the implication of the decomposition of the TM into the MTMs when going to eigenvalues. It is clear we can do singular value decomposition of the TM, however is the highest eigenvalue of the TM then related to the highest eigenvalue of each of the MTMs? This is not a priori clear and can be explained better. In other words, what happens with equation (3) when doing a singular value decomposition?

-- *In contrast to modes, the eigenchannels are defined at a specific frequency. However, the eigenchannels can be inferred from the modal decomposition of the TM. In the localized regime, the modes are generally spectrally isolated and the first transmission eigenchannel correspond to the contribution of the first mode. The first eigenvalue is therefore equal to the strength in transmission of the dominant mode. However, when several modes overlap, the first eigenchannel is built upon interference of several modes and such mapping does not exist. This is now explained in the paragraph below:*

Excitation of specific quasi-normal modes differs from excitation of transmission eigenchannels. The eigenchannels and eigenvalues of the TM can be found via a singular value decomposition in which the TM at a single frequency is expressed as the product of three $N \times N$ matrices, $t(\omega) = UAV^\dagger$. Here A is a diagonal matrix whose elements are the singular values $\sqrt{\tau_i}$, and V and U are unitary matrices and correspond to the waveforms of the transmission eigenchannel on the input and output of the sample, respectively. In contrast to modes, which have a Lorentzian spectrum, the eigenchannels are defined at a specific frequency; a new set of transmission eigenvalues and eigenchannels must be computed at each frequency [5, 22, 53, 54]. However, the spectral characteristics of the channels can be obtained by decomposing the transmission eigenchannels into modes [22, 53]. When a single mode dominates transmission, the MTM for this mode is close to the first eigenchannel. At the resonance, the first transmission eigenvalue is, $\tau_1(\omega_n) = \|W_{Ln}\|^2 \|W_{Rn}\|^2 / (\Gamma_n/2)^2$. When several resonances overlap, however, the first transmission eigenchannel is a combination of modal contributions of several modes.

3. As a minor comment, there are two Equations (4). Also I found the notation of the sum in Eq. (3) somewhat confusing as it looks like a variable Sigma_n among all the other greek variables with subscript n, but probably this is a typesetting issue.

-- *We have now corrected the error in the numbering of Eq. (4) and made the summation sign larger:*

$$t(\omega) = -i \sum_{n=1}^M \frac{W_{Rn} W_{Ln}^T}{\omega - \omega_n + i \frac{\Gamma_n}{2}}$$

4. Is there any limitation or assumption on the dimensionality of the system? It appears the theory is independent of dimensionality, the experiments are in 2D waveguide geometry. What about a slab geometry or open 3D system? For example, in a recent paper on arXiv:1806.01917, it was found that eigenmodes are spatially localized in a slab. Does this affect the expansion of the TM into MTM?

-- *There is no a priori limitation to the geometry or the dimensionality of the system. In arXiv:1806.01917, 2018, the eigenchannels in a slab are indeed transversally localized. However, this does not in-and-of-itself prove that the modes are also localized spatially. The eigenchannels are a superposition of many modes in the diffusion regime and such transverse localization could be due to destructive interference between modes instead of the transverse localization of the modes themselves. However, the eigenchannels are localized as a consequence of the spatial coherence of the wave which is attenuated over distances larger than the transverse spread of the wave due to incoherent diffusion in the slab. But modes are also coherent and may be limited in the transverse direction for the same reason as the eigenchannels are. The point raised is indeed very interesting and bears further investigation.*

5. In Equation (4) (the first one) a field E_{opt} is defined in terms of the transmission matrix and a normalized waveform. It was not entirely clear what was done here. Is E_{opt} supposed to be the optimized field after an optimization of the input state to maximize the total transmission? It seems W_n is the input pattern for only one mode, so it seems that E_{opt} only takes into account the input coupled into one mode n . But then it uses the full transmission matrix so it seems to measure the coupling of an input optimized for mode n , to all other modes. Perhaps this could be explained better in the text, we can work this out from the mathematics but it will help to get some intuition beforehand.

-- *We have rewritten this to make the point clearer that the optimum field is found by maximizing the strength of excitation of an individual mode. This is achieved by exciting the sample with the phase-conjugate of the modal vector found from the MTM associated to the selected resonance. Using the TM matrix, we then study the transmitted waveform corresponding to this excitation. Since the transmitted field is a superposition of several modes, the contribution of other modes also appears in the equation. In general, when several modes overlap, maximizing the strength of a specific mode differs from maximizing the first transmission eigenchannel.*

Modal selectivity. The strength of excitation of an individual mode is maximized when the incoming wave on the left or right excites the sample with the *optimal modal patterns* W_{Ln}^* and W_{Rn}^* , respectively. These are the complex conjugates or the time-reversal of modal speckle patterns at the sample boundaries. Using Eq. (4), the vector of the transmitted field for an excitation of the sample from the left with the normalized optimal waveform, $v_n = W_{Ln}^*/\|W_{Ln}\|$, can be expressed as

$$E_{\text{max}}(\omega) = t(\omega)v_n = -i \frac{\|W_{Ln}\|^2}{\omega - \omega_n + i\frac{\Gamma_n}{2}} \frac{W_{Rn}}{\|W_{Ln}\|} - i \sum_{m \neq n} \frac{W_{Lm}^T W_{Ln}^*}{\omega - \omega_m + i\frac{\Gamma_m}{2}} \frac{W_{Rm}}{\|W_{Ln}\|}. \quad (5)$$

The first term in Eq. (5) gives the contribution of the n^{th} mode to transmission for maximal coupling and the sum in the second term gives the contributions of other modes. Apart from the Lorentzian function, the energy in the mode to which the field is maximally coupled is equal to

$\|W_{Ln}\|^2\|W_{Rn}\|^2$. This can be compared to the average energy for a normalized random incoming waveform v_{rand} which is $\langle |W_{Ln}^T v_{\text{rand}}|^2 \rangle \|W_{Rn}\|^2 = \|W_{Ln}\|^2\|W_{Rn}\|^2/N$. The energy in the mode for maximal coupling in an N -channel system is therefore enhanced by a factor N using the optimal modal pattern. This property is a consequence of the unit-rank of the MTMs. At the same time, the contribution to transmission of the selected mode vanishes for any incoming vector orthogonal to the optimal modal pattern W_{Ln} . Residual transmission is due to the contributions of neighboring modes.

6. In Figure 1(d) they should explain better what are the different colored curves. One of the problems is that the different colors have not been explained in (b) and (c) either, so it is not clear what they represent.

-- The caption of Fig. 1 has now been corrected to make the meaning of the figure clear.

7. In Figures 3 and 4 it appears that the two modes have a very similar pattern over a large part of their volume, while the difference seems to be mainly in the intensity within this pattern. At the point where this was discussed I had a lot of questions about this, which only became more clear when digesting the following theory on Page 10. I think it will help the reader if this similarity is already pointed out in the experimental results, where then can be referred to the theory further on explaining this correlation in more detail.

-- We agree that this will help the reader. We have added a discussion of this point before Fig. 4:

The distributions of energy density for a random wavefront and for the first transmission eigenchannel at a frequency midway between the two resonances $\omega_0 = (\omega_n + \omega_{n+1})/2$, are seen in Fig. 4(a,b) to be primarily mixtures of the modal spatial patterns of the two neighboring modes shown in Fig. 3 [53]. Nevertheless, it is possible to preferentially excite a single mode by adjusting the incident wave to match the pattern of one of the nearby resonant modes. In Fig. 4(c,d), the energy density at the frequency between the modal resonances is shown for maximal coupling to one or the other of the modes. In each case, the energy density matches the spatial distribution of the selected mode shown in Fig. 3a. The degree of modal selectivity between two modes achieved by maximizing the input for one of the modes is reduced as a result of the hybridization and spectral broadening of the modes of the closed system when the sample is coupled to its surroundings. We will see in the theoretical analysis and measurements below that the similarity in modal patterns in the interior of the sample seen in Fig. 3a is a consequence of the bi-orthogonality of the eigenfunctions and the correlation between them.

Reviewer #2

1/ The key mechanism from a microscopic point of view is the excitation of quasi-normal modes by an incident wavefield. It is known that the excitation coefficient of a quasi-normal mode at a given frequency can be expressed as an overlap integral of the incident field with the mode field. See for instance [P. Lalanne et al., Laser & Photonics Reviews 12, 1700113 (2018)] for a recent review on the topic. Considering that resonances of open systems are spectrally broad and their eigenfunctions non-orthogonal, it is then naturally expected that any incident field at a given frequency will also couple to other resonant modes. Supposing that the incident field is

very close to that of the selected mode, the excitation to the other modes should be approximately given by the overlap between these modes and the selected one. Such a concise and simple description of the problem at hand is currently lacking, or is not sufficiently emphasized, giving the wrong impression that the findings of the work should be completely unexpected from the current literature.

Along this line, it is important also to remark that the physics into play here is basically that of coupled resonances, which have been studied quite exhaustively (e.g., for waveguide modes [W. P. Huang, JOSA A 11, 963 (1994)], or more recently for optical resonators [B. Vial and Y. Hao, J. Opt. 18, 115004 (2016)]). The interest in studying disordered systems from the application viewpoint is clear but when it comes to describing the physics of coupled systems, this knowledge base should be acknowledged.

I believe that, in the introduction, the authors should better explain what is expected from existing literature and clarify the novelty of their work (since the authors have, in fact, gone beyond this).

-- The correlation of speckle patterns for modes that are overlapping spectrally, as expressed above, is indeed a fundamental point. It is the crux of our paper, but it is not discussed in the paper by Lalanne et al. In disordered media, the statistics of modes have been studied before, in particular in the pioneering work of Wigner and Dyson. Their work dwelt on the statistics of level spacing and level widths. But this is only part of the story, the full story begins to emerge in our paper which treats the correlation between modal speckle patterns, which is essential for studies of excitation and transport. When modes overlap, the degree of correlation between speckle patterns will indeed limit the modal selectivity, which is explored in this paper. We now reference the physics of modal coupling with cavities, waveguides, or optical resonators and in plasmonics. We also reference the paper by Lalanne et al. and the earlier work by Wigner and by Dyson.

We have now given a broader introduction:

Another approach to controlling propagation within random or structured media might be to manipulate the incident wave to preferentially excite specific quasi-normal modes [23] which have different lifetimes and spatial profiles. Modes of open systems are solutions of the wave equation over the volume of the random medium with outgoing radiation boundary conditions [24-26]. In resonating structures for which the complex eigenvalues and eigenvectors can be found analytically or numerically, the field for any source excitation can be reconstructed from the coherent superposition of modal contributions [26, 27]. Beyond the independent contribution of each mode, the resultant field depends critically upon the interference between the fields of modes that overlap spectrally and spatially. Modal coupling plays a key role in describing the physics of photonic systems such as chaotic cavities [28-32], coupled cavities or waveguides [33, 34], optical resonators [27, 35], quantum plasmonic [36, 37] or disordered media [30, 38-40].

In large complex systems, it is generally not possible to solve for the eigenvectors of the wave equation, but important properties of a system and its coupling to its surroundings can be determined from the statistics of scattering spectra and their analysis into modes or energy levels. Great emphasis has been placed on the probability distributions of level spacings [41-43] and level widths [28, 32, 44]. However, the statistics of level widths and spacings do not directly

yield the statistics of scattering because the scattered wave also reflects the interference between modes and the degree to which modal speckle patterns are correlated.

2/ Following the discussion above, the excitation in the system considered in this work cannot be completely arbitrary; it is driven by a finite number of sources in a system containing a finite number (N) of channels, thereby leading to a lower bound on the spatial field variation that could be achieved for the exciting field. The increase of the mode energy by a factor N compared to the random excitation is a very nice result, but:

- the origin of the factor N is not sufficiently argued and described, in my opinion. It is currently limited to 3 lines (from 98 to 100).
- the experimental proof is limited to a pair of modes (Fig. 2), while one would expect a larger statistics.

The authors should explain in detail where the factor N comes from in their theory and provide additional experimental results on mode energy enhancement with better statistics to be more convincing.

-- *We emphasize that the factor of N is due to the unit-rank of the MTMs. Strong experimental proof is given in Fig. 1d. The ratio of the first and second eigenvalues of an MTM is indeed higher than 10^2 . This strongly supports the unit rank property of the MTM. We then demonstrate in the second part of the following paragraph that due to this unit-rank property, the modal strength is enhanced by a factor N for maximal coupling in comparison to a random illumination. This is highlighted in the discussion below that has been added to the paper.*

Modal selectivity. The strength of excitation of an individual mode is maximized when the incoming wave on the left or right excites the sample with the *optimal modal patterns* W_{Ln}^* and W_{Rn}^* , respectively. These are the complex conjugates or the time-reversal of modal speckle patterns at the sample boundaries. Using Eq. (4), the vector of the transmitted field for an excitation of the sample from the left with the normalized optimal waveform, $v_n = W_{Ln}^*/\|W_{Ln}\|$, can be expressed as

$$E_{\max}(\omega) = t(\omega)v_n = -i \frac{\|W_{Ln}\|^2}{\omega - \omega_n + i\frac{\Gamma_n}{2}} \frac{W_{Rn}}{\|W_{Ln}\|} - i \sum_{m \neq n} \frac{W_{Lm}^T W_{Ln}^*}{\omega - \omega_m + i\frac{\Gamma_m}{2}} \frac{W_{Rm}}{\|W_{Ln}\|}. \quad (5)$$

The first term in Eq. (5) gives the contribution of the n^{th} mode to transmission for maximal coupling and the sum in the second term gives the contributions of other modes. Apart from the Lorentzian function, the energy in the mode to which the field is maximally coupled is equal to $\|W_{Ln}\|^2 \|W_{Rn}\|^2$. This can be compared to the average energy for a normalized random incoming waveform v_{rand} which is $\langle |W_{Ln}^T v_{rand}|^2 \rangle \|W_{Rn}\|^2 = \|W_{Ln}\|^2 \|W_{Rn}\|^2 / N$. The energy in the mode for optimal coupling in an N -channel system is therefore enhanced by a factor N using the optimal modal pattern. This property is a consequence of the unit-rank of the MTMs. At the same time, the contribution to transmission of the selected mode vanishes for any incoming vector orthogonal to the optimal modal pattern W_{Ln} . Residual transmission is due to the contributions of neighboring modes.

3/ The title and abstract suggest that the authors provide a general approach to the selective excitation of quasi-normal modes in open disordered systems. The theory that is developed here is however approximate, in the sense that leakage to the continuum is treated as a

perturbation on a closed system. It is clear that the theoretical treatment will progressively fail for systems with lower quality factor resonators.

-- *It is in the nature of a short title that it catches the key point of the work in just a few words. The title, “Selectively exciting quasi-normal modes in open disordered systems” does not catch every nuance, but it does catch the main idea.*

I believe that the authors should be extremely clear on this aspect in the manuscript.

-- *The view that the modal approach follows from perturbation theory is also expressed in section 6.2 of the paper by Lalanne et al., Laser & Photonics Reviews 12, 1700113 (2018)], where the following is written: ” The perturbation description is valid in the limit of large Q 's only. With increasing energy dissipation, either by leakage or absorption, the resonance peaks broaden and start overlapping, as sketched in Figure 11c. In the spectral regions where the overlap is important, several modes contribute to the LDOS and Fano resonances may arise from the interference between different modes. Because of these interferences, the LDOS can no longer be seen as a sum of Lorentzian positive contributions.” It seems that the discussion here of the breakdown of the modal picture relates to the transmission no longer being a sum of Lorentzians once modes begin to overlap. However, the field may be given by a sum of spectra with the frequency given in our Eq. (1),*

$$\frac{t_{ba}^n}{\omega - \omega_n + i\Gamma_n/2}$$

which exhibits interference as opposed to the sum of Lorentzian positive contributions, each of which is real and positive. Thus the modal picture is described as failing here not because of an inherent breakdown modal analysis but because of interference.

The notion of modes is inherent in the work of Wigner and Dyson and also in the discussion in a recent article by Alpegiani et al. on Quasinormal-Mode Expansion of the Scattering Matrix, Phys. Rev. X 7, 021035 (2017). This paper addresses the reconstruction of the S matrix from the eigenfunctions in nanophotonic systems with multiple overlapping modes. In this paper, the authors develop a general expansion of the scattering into quasi-normal modes and demonstrate that the scattering matrix can be reconstructed from the eigenfrequencies and the far-field properties of the eigenfunctions. The modal coefficients are built upon contributions of all modes through a coupling matrix (see Eq. (16) of this article). Using this decomposition, the MTMs are still of unit-rank. This suggests that our approach should remain valid even when the modal overlap increases. We mention that measurements are carried out in a regime of high Q factor in the text and discuss this point in the following paragraph in the Discussion section:

This investigation of selective excitation of quasi-normal modes in disordered system has been carried out in systems with resonances with high quality factors. The decomposition of the TM into MTMs is more challenging for diffusive waves when the degree of modal overlap is high. However, as Alpegiani et al. have demonstrated analytically, the scattering matrix can be reconstructed from the far-field properties of the eigenmodes for any degree of overlap [26]. The modal coefficients depend on all other contributing modes through a coupling matrix, but MTMs are still of unit rank. Future studies will be dedicated to exploring the characteristics of modes and the limits of selectivity in systems with strong modal overlap. This will advance a more

comprehensive understanding of the relationship between eigenchannels, time-delay eigenstates and quasi-normal modes of open system.

The generalization of their results to arbitrary systems could be discussed in the Discussion section.

We have added a discussion of the application of our results to random lasing in the next to last paragraph of the Discussion section. We now show that a proper understanding of the regime of moderate overlap in diffusive samples is essential for understanding random lasing. We show that when a single or a few modes are excited the modal character of the excitation is key. Since modes in open diffusive systems are extended, energy can be deposited into the middle of the sample and this leads to low threshold lasing. In the last paragraph we discuss the generalization of the results to arbitrary systems in the last paragraph:

Selecting specific modes in samples in which the wave is localized would make it possible to deliver energy to specific regions of a sample. In the case of moderate modal overlap in open random systems, the modes extend over the entire sample, even in absorbing samples, and so if a single mode or a small number of modes is selected, it would be possible to deliver energy to the center of the sample. In contrast, when many modes overlap, the average profile of energy density within the sample is determined by the diffusion equation and is concentrated within an absorption depth of the sample $L_a = (D\tau_a)^{1/2}$, where D is the diffusion coefficient and τ_a is the absorption time, and particularly the absorption associated with exciting gain in the medium [63]. As a result, light emitted from excited samples with weakly overlapping modes, in which energy penetrates more deeply into a random medium, is longer lived than in samples in with stronger modal overlap. There is therefore greater opportunity for the emitted photons to stimulate emission before escaping the sample. The lasing threshold will consequently be lowered and narrow-line emission will be observed [64]. In general the smaller the degree of modal overlap, the more it is possible to deposit energy into a random absorbing medium. Finding the MTM and then pumping from the front of the sample [64, 65] could thereby lower the threshold of random lasers via coherent feedback.

Modal decomposition of the TM has been demonstrated with microwave radiation but is in principle possible in optics. Measurement of spectrally resolved TM has indeed been reported recently [66]. Modal selectivity could then be utilized to enhance light-matter interactions in photonic materials [67], solar cells [68] or biomedical optics [69]. The use of MTMs is not restricted to random media but can also be applied to structured media such as optical microcavities [31], or photonic crystals [70].

4/ In the work, the authors support their various, successive claims by (i) experiments, (ii) finite-element simulations and (iii) recursive Green's function simulations on different system. It is not clear why the authors have chosen not to demonstrate all claims experimentally. The oscillation between various theories, an experiment and different numerical methods complicates the understanding of the work, especially when the choices made are not explained.

-- *We have now removed the finite-element (COMSOL) simulations from the main manuscript. The result concerning the hybridization of two modes can actually be supported by measurements as we have it now. The COMSOL simulations are now included in the*

Supplementary Information. However, the last figure and the Green's function simulations are needed because a number of samples must be studied to explore statistical questions

MINOR POINTS:

5/ Details on the experiment are given in the main text - not in the Methods - and the numerical analysis and numerical methods are in general not sufficiently described. They deserve sections in the Methods. In particular, more details about the reconstruction of the spatial energy distribution (lines 180-182) and the COMSOL simulations (which boundary conditions? which solver?) are needed.

-- *We provide more details about the experimental setup and the reconstruction of the spatial energy distribution in the Method section. We have add the following section:*

Experimental Setup. The aluminum cavity has length $L = 500$ mm, width $W = 268$ mm and height $H = 8$ mm and only supports a single waveguide mode in the vertical dimension over the frequency range of the measurements. The antennas are waveguide to coax adaptors designed for the Ku band (12-18 GHz). Measurements of individual elements of the TM between two antenna arrays are made with use of electro-mechanical switches and a vector network analyzer. The channels of the switches that are turned off are matched to a 50Ω load so that the boundary conditions of the system do not change when different emitting source and receiving antennas are used. The spacing between two antennas on the left and right sides of the sample is metallic as seen in Fig. 1a.

The field inside the waveguide is detected by inserting an antenna sequentially into a square grid of holes which are 4 mm in diameter and spaced by 8 mm on a side. The transmission coefficient $e_a(x, y, \omega)$ is measured between each source antenna and a wire antenna inserted 0.5 mm below the bottom of the 6-mm thick aluminum cover. The penetration depth of the antenna is small enough that it does not distort the field profile in the waveguide. The spatial energy distribution $I(x, y)$ for any incoming vector v is then reconstructed from the coherent superposition of the fields arising from each source antenna, $I(x, y) = |\sum_a e_a(x, y) v_a|^2$. Once the decomposition into modes of the field inside the sample, $e_a(x, y, \omega) = \sum_n e_a^n(x, y) \varphi_n(\omega)$, has been obtained, the modal spatial profile is found from an average on incoming channels $I_n(x, y) = \langle |e_a^n(x, y)|^2 \rangle$. We also present in Fig. 2f the modal energy density profile for incoming vector v computed from $W_n(x, y) = |\sum_a e_a^n(x, y) v_a|^2$. For maximal modal coupling $v = W_{Ln} / \|W_{Ln}\|$ and for vanishing coupling v is orthogonal to W_{Ln} .

The COMSOL simulations are now included in the Supplementary Information, where we have added details about the boundary conditions that are used to compute the transmission matrix and find the eigenmodes.

6/ The figures are not excellent. There are no scales on the field maps, no description of the panels ("random wavefront", etc) in Fig. 3, no legend on Fig. 5, ... The frequencies of the modes in Fig. 2 do not seem to be 11.434 GHz and 11.461 GHz as stated in the text. Adding the variables (t_{ba} , δ , ...) on the figure axis titles would help.

-- *This has now been corrected.*

7/ Some text is missing on line 302.

-- *This has now been corrected.*

Reviewers' Comments:

Reviewer #1:

Remarks to the Author:

The authors have carefully addressed the comments raised by both reviewers. From the comments it is clear that the other reviewer has a more deep theoretical knowledge and should have a more decisive vote on the novelty of the theoretical concept. From a non-expert point of view, it is my opinion that the work has been improved in its accessibility for a general audience.

Regarding novelty and suitability for Nature Communications, I believe that the demonstration of modal decomposition of the transmission matrix is conceptually novel and of interest for a wide community of people working in complex media and particularly wavefront control. I agree with the other reviewer that perhaps some aspects could have been expected, in particular the loss of specificity in coupling when modes are spectrally overlapping. I agree with the authors that the microscopic picture so far has not been well addressed, and the direct visualization of non-orthogonality in an experiment is appreciated as a novel result.

Regarding the upgrading of figures to experimental results, this certainly helps the value of the paper. I would recommend that each of the figures explicitly mentions that these are experimental maps, because it is very hard to see this by eye and sometimes we may think these are simulations while in fact they are experiments.

So altogether it is my opinion that the current manuscript is fine and should eventually be published. I maintain my original opinion that the topic is of interest and level of novelty suitable for Nature Communications.

Reviewer #2:

Remarks to the Author:

Review of the revised manuscript entitled "Selectively exciting quasi-normal modes in open disordered systems" by Matthieu Davy and Azriel Z. Genack.

In my previous report, I asked to clarify the generality and originality of the authors' findings, provide a more thorough description and/or stronger experimental or numerical evidence of certain conclusions, and improve the readability of the text and quality of the figures. The authors have replied to all points in their letter and made substantial modifications to the manuscript, clarifying the generality and originality of their work by placing it in a proper context, making it more accessible to a broader readership by better introducing the key concepts and moving some simulation results to SI, providing finally more information on the methods they have used.

The manuscript has been significantly improved, in my opinion, and I am happy to recommend it for publication in Nature Communications.

To clarify one of the points I made previously regarding the applicability of the approach for low quality factor resonances, my observation was that (i) the theoretical formalism describes the problem in terms of the modes of the closed system that are leaking to the continuum (are Eqs. 2, 3, 9, ... valid for systems with low-Q resonances?), and (ii) the assumption of high-quality factor resonances is needed to be able to write that the coupling vectors are frequency-independent. Nevertheless, the authors are sufficiently clear in their manuscript on this aspect.